# Effect of disclosure of HIV status on patient representation and adherence to clinic visits in eastern Uganda: A propensity-score matched analysis

Jonathan Izudi[1,2,3]*, Stephen Okoboi[2,3], Paul Lwevola[2], Damazo Kadengye[4], Francis Bajunirwe[1]

1 Department of Community Health, Faculty of Medicine, Mbarara University of Science and Technology, Mbarara, Uganda, 2 Institute of Public Health and Management, Clarke International University, Kampala, Uganda, 3 Infectious Diseases Institute, School of Medicine, Makerere University College of Health Sciences, Kampala, Uganda, 4 African Population and Health Research Center (APHRC), APHRC Campus, Nairobi, Kenya

* jonahzd@gmail.com

## Abstract

**Data Availability Statement:** All relevant data are within the manuscript and its Supporting information files.

### Background

Disclosure of human immunodeficiency virus (HIV) status improves adherence to antiretroviral therapy (ART) and increases the chance of virological suppression and retention in care. However, information on the effect of disclosure of HIV status on adherence to clinic visits and patient representation is limited. We evaluated the effects of disclosure of HIV status on adherence to clinic visits and patient representation among people living with HIV in eastern Uganda.

### Methods

In this quasi-randomized study, we performed a propensity-score-matched analysis on observational data collected between October 2018 and September 2019 from a large ART clinic in eastern Uganda. We matched participants with disclosed HIV status to those with undisclosed HIV status based on similar propensity scores in a 1:1 ratio using the nearest neighbor caliper matching technique. The primary outcomes were patient representation (the tendency for patients to have other people pick-up their medications) and adherence to clinic visits. We fitted a logistic regression to estimate the effects of disclosure of HIV status, reported using the odds ratio (OR) and 95% confidence interval (CI).

### Results

Of 957 participants, 500 were matched. In propensity-score matched analysis, disclosure of HIV status significantly impacts adherence to clinic visits (OR = 1.63; 95% CI, 1.13–2.36) and reduced patient representation (OR = O.49; 95% CI, 0.32–0.76). Sensitivity analysis showed robustness to unmeasured confounders (Gamma value = 2.2, $p$ = 0.04).

**Funding:** The author(s) received no specific funding for this work.

**Competing interests:** The authors have declared that no competing interests exist.

## Conclusions

Disclosure of HIV status is associated with increased adherence to clinic visits and lower representation to collect medicines at the clinic. Disclosure of HIV status should be encouraged to enhance continuity of care among people living with HIV.

## Introduction

Globally, Human Immunodeficiency Virus (HIV) remains a major public health problem and has so far claimed 35 million lives. At the end of 2019, approximately 38 million people were living with HIV and nearly 1.7 million become newly infected with HIV [1]. Of all the regions in the world, the African region is the most affected by HIV, with at least 25 million people living with HIV besides being home to almost two-thirds of the global new HIV infections [2]. In Uganda, estimates indicated that 1.4 million people were living with HIV at the end of 2019, and of this, 88% knew their HIV status while 87% of those who knew their HIV status are on HIV treatment [3]. Data about disclosure of HIV status at the national level is lacking but observational studies conducted in recent years and different populations indicate that at least 80% of people living with HIV have disclosed their HIV status to someone [4–6].

Disclosure of HIV status involves revealing one's HIV positive status to a sexual partner(s), family members, or others in their social circle [7], and is considered a key component of the positive health, dignity, and prevention (PHDP) package within Uganda's HIV programming. The package offers an option for a provider and/or counselor-mediated or supported disclosure for people who are having difficulty disclosing their HIV status [3, 8]. The disclosure of HIV status has several benefits namely, improved adherence to medications, access to essential services, reduced psychological distress, increased likelihood of appropriate disclosure to other people, better engagement in HIV-related care, improved understanding of HIV and related conditions, and enhanced uptake of the PHDP package [8], improved quality of life, better immune recovery as reflected by rising CD4 cell counts, and viral load suppression [9, 10]. Also, disclosure of HIV status is associated with a higher likelihood of retention [11] while non-disclosure of HIV status is associated with an increased risk of loss to follow-up [12].

Furthermore, disclosure of HIV status is associated with a higher likelihood of condom use, an increased social support, and knowledge of the partner's HIV status [7].

Adherence to HIV clinic visits has several benefits such as a lower risk of mortality [13], adherence to medication, slower progression of disease, higher odds of viral load suppression and immune recovery [14, 15], and lower risk of hospitalization [16]. In situations where a patient is not able to attend a clinic visit in person, they may send a representative to collect HIV medications for them, and this patient representation is an allowable practice in Uganda. Patient representation is an alternative to individual clinic visits and a recognized form of clinic attendance. Although acceptable, patient representation may result in loss of benefits of individual clinic attendance namely, the provision of ongoing counseling and support, and missed clinical, immunologic, and virologic monitoring which are intended to monitor treatment success. Numerous benefits of disclosure of HIV status have been described, but there are limited studies on the effect of disclosure of HIV status on patient representation and adherence to clinic visits. We hypothesized that disclosure of HIV status is associated with lower patient representation and higher adherence to clinic visits [17]. Therefore, the primary objective of this study was to assess the effect of disclosure of HIV status on patient representation and adherence to clinic visits among people living with HIV in an ART clinic in eastern Uganda.

## Methods and materials

### Data source and study setting

The data (S1 File) for this study were drawn from the routine health care records of the HIV clinic at Kidera Health Center HC IV, the largest ART clinic in Buyende district in rural eastern Uganda.

The population of the district is 323, 067 people of which 50.9% (164,452) are females [18]. The health facility serves as the referral site for HIV care in the district, serving about 32% of people living with HIV in the district. The health facility has a catchment population of approximately 60,000 people. Besides providing comprehensive HIV care, the health facility provides promotion, preventive, curative, and rehabilitative health services to the catchment population.

### Study population

The study population consisted of a census of people living with HIV started on ART between October 2018 and September 2019. The eligible participants included those aged ≥15 years and enrolled in care for ≥6 months during the review period. We excluded participants transferred to other health facilities because it was logistically infeasible to follow all of them and obtain data about their HIV disclosure status. We also excluded participants who were documented dead to prevent a biased estimate of the HIV disclosure effect. Further, we excluded participants whose disclosure of HIV status had occurred after the study outcomes as this would result into an inaccurate measure of the temporal relationship between disclosure of HIV status and the study outcomes.

### The operation of the ART clinic

The ART clinic is run by a clinical officer, two nurses, a counselor, and two volunteers who provide clinical care, nursing care, psychosocial counseling support, and health education to people living with HIV. The clinic runs twice a week and has special clinics for children and adolescents.

Patients who are stable on ART namely those, 1) who have been on their current ART regimen for more than a year; 2) with undetectable viral load at the most recent test defined as viral load less than 1000 copies per ml in the last 12 months; 3) in the World Health Organization (WHO) clinical stages I and II; and, 4) who have demonstrated good adherence defined as more than 95% ART adherence in the last six consecutive months [3], receive refills of anti-retroviral drugs (ARVs) to last 3–6 months. The rest of the patients are considered unstable and receive refills of anti-retroviral drugs (ARVs) to last one month.

To track adherence to clinic visits, the ART clinic maintains an appointment register where the scheduled clinic visit dates of all patients are captured. Although the study participants have different ART regimens, the clinic uses an appointment system that enables multi-month dispensing of drugs, usually is 1–2 months of refill. Adherence to the clinic visit is updated in real-time on clinic days by a records assistant. There is also a register for the missed appointment to record all patients who miss an appointment to enable follow-up either immediately through phone calls or home visits within 2–5 days.

To minimize non-adherence to clinic visits, reminders are sent before scheduled clinic visits, and for those who have failed to come to the ART clinic as scheduled, the ARV refills are done in the community. Once the ARV refills are completed, the relevant clinic registers are updated. For patients who fail to adhere to scheduled clinic visits, their HIV medications can be collected by a representative who in most cases is a treatment supporter. However, not

more than 2 consecutive representations are allowed. Following the Uganda Ministry of Health HIV treatment guidelines [3], a patient is considered to have dropped from HIV care if he/she is lost for at least 3 months and 3 attempts to follow-up have been unsuccessful.

### Data abstraction

We abstracted data from the ART register for all eligible patients (patients started on ART between October 2018 and September 2019, aged ≥15 years, and enrolled in care for ≥6 months during the review period) using a standardized data abstraction tool. We received a waiver of informed consent from Clarke International University Research Ethics Committee (CIU-REC) to retrieve and analyze patient records since it would be impossible and logistically inefficient to reach all the patients (CIU-REC number CLARKE-2020-16).

### Study design

To measure the effects of an intervention, a randomized control trial (RCT) is the gold standard because randomization achieves comparability by balancing both measured and unmeasured participant characteristics across the intervention and treatment groups [19]. However, an RCT is not always feasible or ethical for interventions that are known to have certain positive benefits such as disclosure of HIV status. We, therefore, used observational data to approximate an RCT by creating two groups, distinguished by disclosure status (the exposure factor) but similar based on observed covariates, achieved using propensity-score matching [20]. Propensity-score matched analysis is a statistical approach that simulates an RCT by balancing all observed confounders or covariates across treatment groups except the treatment [21]. Since this analytic approach does not rely on true randomization, the study design is quasi-randomized [20].

### Measurements: Exposure and outcomes

The exposure in this study was the disclosure of HIV status measured as a dichotomous variable at the second visit (week 2) after initiation of ART and updated as treatment progresses.

Disclosure of HIV status was defined as revealing HIV positive status to a sexual partner(s), family members, or others in their social circle [7]. The exposed group consisted of participants with disclosed HIV status while the unexposed group comprised of participants with undisclosed HIV status. Our analysis considered the disclosure of HIV status that preceded the study outcomes to ensure a valid measure of effect. The study outcomes were patient representation and adherence to clinic visits in the past 6 months, measured following the Uganda Ministry of Health guidelines [22].

- **Patient representation:** This was measured as the practice where the individual patient does not show up at the HIV clinic on the scheduled date but delegates someone to pick up the medications for them. Participants who did not show up at the HIV clinic on one or more occasions in the past 6 months were considered to have been represented. This outcome was measure as a binary variable (yes or no).

- **Adherence to clinic visits**: This was a binary variable (yes or no) measured as adherence to the visit at the HIV clinic where the individual patient attends clinic on the date that he/she was scheduled or within seven days before or after the scheduled date. All participants who did not adhere to their scheduled visit within the recommended period (±7 days) were considered non-adherent to the clinic visits.

- **Matching covariates:** These included age in years (≤24, 25–50, and >50), sex (female or male), marital status (single, married, and separated), level of education (none, primary, secondary, tertiary, and above), availability of a source of income (no or yes), estimated distance from home to a health facility in kilometers (<5, 5–10, >10), ease of access to health facility (no or yes), current alcohol consumption (no or yes), current smoking status (no or yes), duration on ART in years (<5, 5–9, >10), receipt of pre-ART counseling (no or yes), receipt of pre-tuberculosis preventive therapy counseling (no or yes), and experience of tuberculosis preventive therapy-related side effects (no or yes).

## Statistical analysis

The analysis was performed in R statistical software and programming language [23], using the *MatchIt* [24] and *tableone* packages [25]. We performed a descriptive analysis to summarize categorical data like sex using frequencies and percentages and numerical data like age using the mean with standard deviations or median with interquartile range.

In the propensity score-matched analysis, we selected 12 matching covariates as already described. These covariates were selected because they are known to be associated with the study outcomes and exposure thus preserving the assumption of unconfoundedness of the association between the exposure and the outcome(s) [26, 27]. Under this assumption, treatment assignment and participant responses are conditionally independent after controlling for covariate effects that determine the assignment mechanism [28].

We generated propensity scores by regressing the matching covariates on disclosure of HIV status in a logit model and assessed the initial balance in propensity scores between the groups using a back-to-back histogram [29]. We then matched participants with undisclosed HIV status (the unexposed group) to participants with disclosed HIV status (the exposed group) on similar propensity scores [24]. We employed several matching approaches. Briefly, we used greedy matching approaches namely nearest neighbor matching with and without caliper adjustment [27]. Caliper is the distance within which the matches were considered. In nearest neighbor matching without caliper adjustment, one participant in the HIV disclosed group was selected at random and matched to one participant in the undisclosed HIV status with the closest propensity score. In nearest neighbor matching with caliper adjustment, the matching was performed within a caliper of 20% of the standard deviation of the propensity score to prevent bias from distant matches. The matching was performed without replacement.

We also employed optimal matching namely optimal pair matching and optimal full matching [28]. In optimal pair matching, we matched the participants in pairs and removed the unmatched pairs from the analysis. In optimal full matching, participants with disclosed HIV status were matched to those with undisclosed HIV status in the ratio of 1: many or many: 1. In addition, we performed exact matching where the participants were matched on the same value of propensity score [30]. We considered the best matching approach as one that balanced all the covariates between the disclosed HIV status and the undisclosed HIV status groups.

We assessed covariate balance using standardized mean differences (SMD) and considered an SMD less than 0.2 to confirm covariate balance [31]. We graphically explored covariate balance using a jitter plot and histogram, and we considered the distributional similarity of propensity scores as confirmatory of covariate balance [31, 32]. We saved the matched dataset for the outcome analysis.

In the unmatched and matched datasets, we separately performed outcome analysis with logistic regression fitted using the generalized linear model with a logit-link and binomial family and reported the results using the odds ratio (OR) and the corresponding 95% confidence

interval (CI). The odds ratio indicates the measure of average treatment effect on the treated (ATT), a measure of the effect of HIV status disclosure for those with disclosed HIV status. We performed a sensitivity analysis using Rosenbaum Wilcoxon's signed-rank test to check the robustness of our results to hidden bias [29] and assessed this by examining the p-values that there is no hidden bias with varying values of the sensitivity parameter gamma.

## Results

### Study profile and matching technique

The dataset comprised 959 patients. Fig 1 summarizes our study profile and shows that of the 959 participants, 293 (30.6%) had disclosed HIV status in the unmatched data. We matched 500 participants in the ratio of 1:1 using the nearest neighbor caliper matching, representing 52.1% of the original data. The caliper used was 0.0265, which was the 20% of the standard deviation of the propensity score.

### Covariate balance before and after propensity-score matching

In the unmatched cohort data (Table 1), we observed systematic differences among participants with disclosed HIV status and undisclosed HIV status regarding the covariates of sex, marital status, and level of education, source of income, and distance and accessibility to a health facility.

Age, alcohol consumption, smoking, duration of ART, pre-ART, and pre- of tuberculosis preventive therapy counseling, and experience of tuberculosis preventive therapy-related side effects were the only characteristics that showed comparable distribution between participants

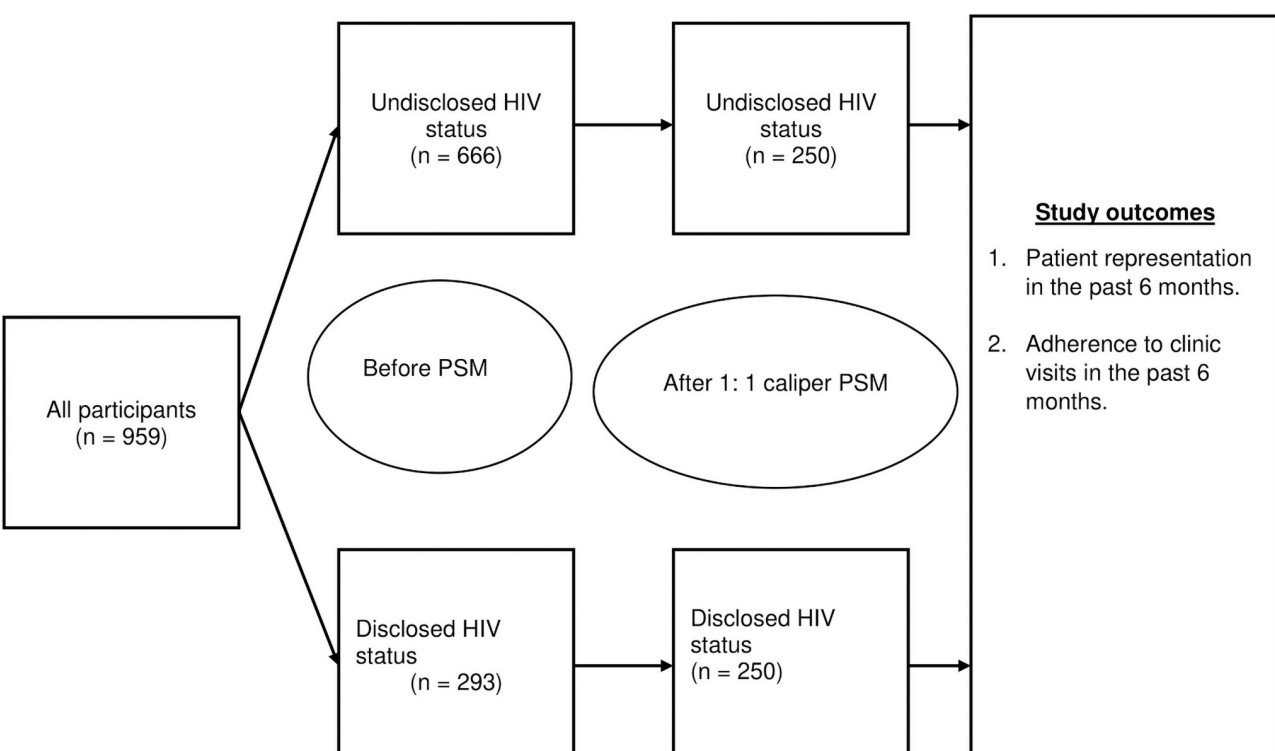

**Fig 1. Study profile of HIV status disclosure before and after propensity score matching.**

**Table 1. Distribution of participants' characteristics before and after propensity-score matching.**

| Variable | Level | Original (Unmatched) cohort | | | | PSM matched cohort | | | |
| | | HIV status disclosure | | | | HIV status disclosure | | | |
| | Sample size | All, | No, | Yes, | SMD | All, | No, | Yes, | SMD |
| | | n = 959 (%) | n = 666 (%) | n = 293 (%) | | n = 500 (%) | n = 250 (%) | n = 250 (%) | |
| Age categories (years) | ≤24 | 104 (10.8) | 73 (11.0) | 31 (10.6) | 0.023 | 58 (11.6) | 30 (12.0) | 28 (11.2) | 0.059 |
| | 25–50 | 631 (65.8) | 436 (65.5) | 195 (66.6) | | 327 (65.4) | 160 (64.0) | 167 (66.8) | |
| | >50 | 224 (23.4) | 157 (23.6) | 67 (22.9) | | 115 (23.0) | 60 (24.0) | 55 (22.0) | |
| | Mean (SD) | 41.1 (13.8) | 41.2 (13.8) | 40.9 (13.7) | 0.026 | 40.7 (14.2) | 40.6 (14.4) | 40.8 (14.1) | 0.009 |
| Sex | Female | 561 (58.5) | 400 (60.1) | 161 (54.9) | 0.104 | 276 (55.2) | 135 (54.0) | 141 (56.4) | 0.048 |
| | Male | 398 (41.5) | 266 (39.9) | 132 (45.1) | | 224 (44.8) | 115 (46.0) | 109 (43.6) | |
| Marital status | Single | 211 (22.0) | 165 (24.8) | 46 (15.7) | 0.255 * | 94 (18.8) | 50 (20.0) | 44 (17.6) | 0.072 |
| | Married | 663 (69.1) | 451 (67.7) | 212 (72.4) | | 358 (71.6) | 175 (70.0) | 183 (73.2) | |
| | Separated | 85 (8.9) | 50 (7.5) | 35 (11.9) | | 48 (9.6) | 25 (10.0) | 23 (9.2) | |
| Level of education | None | 423 (44.1) | 293 (44.0) | 130 (44.4) | 0.340* | 228 (45.6) | 115 (46.0) | 113 (45.2) | 0.138 |
| | Primary | 297 (31.0) | 200 (30.0) | 97 (33.1) | | 160 (32.0) | 80 (32.0) | 80 (32.0) | |
| | Secondary | 121 (12.6) | 72 (10.8) | 49 (16.7) | | 71 (14.2) | 31 (12.4) | 40 (16.0) | |
| | Tertiary and above | 118 (12.3) | 101 (15.2) | 17 (5.8) | | 41 (8.2) | 24 (9.6) | 17 (6.8) | |
| Has a source of income | No | 604 (63.0) | 396 (59.5) | 208 (71.0) | 0.244 * | 341 (68.2) | 169 (67.6) | 172 (68.8) | 0.026 |
| | Yes | 355 (37.0) | 270 (40.5) | 85 (29.0) | | 159 (31.8) | 81 (32.4) | 78 (31.2) | |
| Distance to health facility (km) | <5 | 105 (10.9) | 65 (9.8) | 40 (13.7) | 0.333* | 205 (41.0) | 103 (41.2) | 102 (40.8) | 0.081 |
| | 5–10 | 316 (33.0) | 194 (29.1) | 122 (41.6) | | 60 (12.0) | 33 (13.2) | 27 (10.8) | |
| | >10 | 538 (56.1) | 407 (61.1) | 131 (44.7) | | 235 (47.0) | 114 (45.6) | 121 (48.4) | |
| Accessible health facility | No | 787 (82.1) | 533 (80.0) | 254 (86.7) | 0.180 | 416 (83.2) | 203 (81.2) | 213 (85.2) | 0.107 |
| | Yes | 172 (17.9) | 133 (20.0) | 39 (13.3) | | 84 (16.8) | 47 (18.8) | 37 (14.8) | |
| Drinks alcohol | No | 951 (99.2) | 663 (99.5) | 288 (98.3) | 0.122 | 498 (99.6) | 248 (99.2) | 250 (100.0) | 0.127 |
| | Yes | 8 (0.8) | 3 (0.5) | 5 (1.7) | | 2 (0.4) | 2 (0.8) | 0 (0.0) | |
| Smokes cigarettes | No | 958 (99.9) | 666 (100.0) | 292 (99.7) | 0.083 | 500 (100.0) | 250 (100.0) | 250 (100.0) | <0.001 |
| | Yes | 1 (0.1) | 0 (0.0) | 1 (0.3) | | | | | |
| Duration on ART (years) | <5 | 579 (60.4) | 405 (60.8) | 174 (59.4) | 0.056 | 191 (38.2) | 99 (39.6) | 92 (36.8) | 0.146 |
| | 5–9 | 362 (37.7) | 250 (37.5) | 112 (38.2) | | 301 (60.2) | 145 (58.0) | 156 (62.4) | |
| | >10 | 18 (1.9) | 11 (1.7) | 7 (2.4) | | 8 (1.6) | 6 (2.4) | 2 (0.8) | |
| Received Pre-ART counseling | No | 13 (1.4) | 10 (1.5) | 3 (1.0) | 0.043 | 7 (1.4) | 4 (1.6) | 3 (1.2) | 0.034 |
| | Yes | 946 (98.6) | 656 (98.5) | 290 (99.0) | | 493 (98.6) | 246 (98.4) | 247 (98.8) | |
| Received pre-TPT counseling | No | 48 (5.0) | 33 (5.0) | 15 (5.1) | 0.008 | 27 (5.4) | 16 (6.4) | 11 (4.4) | 0.089 |
| | Yes | 911 (95.0) | 633 (95.0) | 278 (94.9) | | 473 (94.6) | 234 (93.6) | 239 (95.6) | |
| Experienced TPT side effects | No | 951 (99.2) | 659 (98.9) | 292 (99.7) | 0.085 | 499 (99.8) | 250 (100.0) | 249 (99.6) | 0.09 |
| | Yes | 8 (0.8) | 7 (1.1) | 1 (0.3) | | 1 (0.2) | 0 (0.0) | 1 (0.4) | |

TPT: Tuberculosis preventive therapy; ART: Antiretroviral therapy; SMD: Standardized mean difference;

*denotes covariate imbalance with SMD>0.2.

with and without disclosed HIV status (Table 1). We achieved comparable distribution of all these variables in the propensity-score matched data between participants with and without disclosed HIV status (all, SMD<0.2).

Figs 2 and 3 show the distributional similarity of these covariates. Fig 2 (left-hand side) shows that in the unmatched data (raw treated versus raw control), the distribution of the propensity-scores across the disclosure of HIV status was not uniform suggesting an imbalance in covariates. However, after propensity-score matching (matched treated versus matched

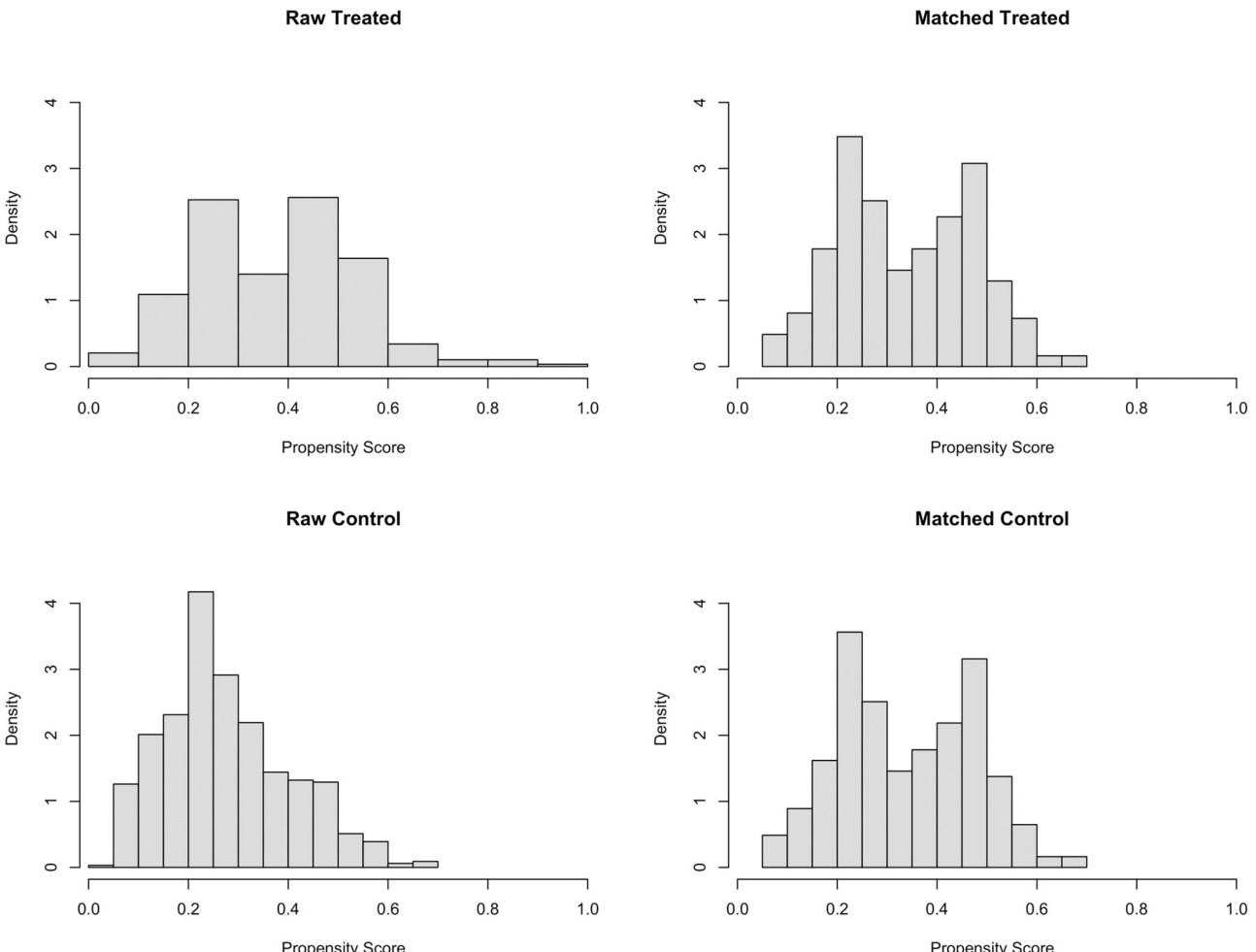

**Fig 2. The histograms show the distribution of propensity scores before and after matching for the treated (disclosed HIV status) and untreated (undisclosed HIV status) groups.** Raw treated: unmatched disclosed HIV status group; Raw control: Unmatched undisclosed HIV status group; Matched treated: matched disclosed HIV status group; Matched control: matched undisclosed HIV status group.

control), the right-hand side of Fig 2, the propensity-scores are similarly distributed across the disclosure of HIV status thus confirming the covariates are balanced. Fig 3 is a jitter plot showing the distribution of propensity scores before and after matching in both groups of participants. The middle section enclosed by a box shows that the treated (disclosed HIV status) and untreated (undisclosed HIV status) groups are similar after matching compared to before matching.

## Effect of disclosure of HIV status on adherence to clinic visits and patient representation

**Unadjusted and adjusted outcome analysis.** In the unmatched data (Table 2), the results show that adherence to clinic visits in the past 6 months was higher among those who disclosed compared to those with undisclosed HIV status: 129 (44.0%) versus 155 (23.3%), respectively ($p<0.001$). The regression analysis showed disclosure of HIV status was associated with an increased likelihood of adherence to clinic visits in the unadjusted (Unadjusted OR (uOR) = 2.59; 95% CI, 1.94–3.48) and adjusted analysis (Adjusted OR (aOR) = 2.04; 95% CI, 1.43–

**Distribution of Propensity Scores**

**Fig 3. A jitter plot showing the distribution of the propensity scores before and after matching.** Raw treated: unmatched disclosed HIV status group; Raw control: Unmatched undisclosed HIV status group; Matched treated: matched disclosed HIV status group; Matched control: matched undisclosed HIV status group.

2.91). Patient representation was lower among participants with disclosed HIV status compared to those with undisclosed HIV status: 212 (72.4%) versus 534 (80.2%), respectively (p = 0.009). Disclosure of HIV status was associated with a 35% reduction in patient representation at unadjusted analysis (uOR = 0.65; 95% CI, 0.47–0.89) and by 45% at adjusted analysis (aOR = 0.55; 95% CI, 0.39–0.79).

**Propensity-score matched outcome analysis.** In the propensity-score matched data (Table 2), the analysis showed that disclosure of HIV status is associated with higher odds of adherence to clinic visits (OR = 1.63; 95% CI, 1.13–2.36). Disclosure is associated with a more than 50% reduction in the odds of patient representation (OR = O.49; 95% CI, 0.32–0.76).

## Sensitivity analysis results

The sensitivity analysis results showed that in the presence of hidden bias and confounding, the unconfounded estimate was 0.3815 corresponding to a gamma value of 1. A gamma value

**Table 2. The effect of disclosure of HIV status on adherence to clinic visits and patient representation in the past 6 months.**

| | | Original (unmatched) cohort | | | | PSM matched cohort | | |
| | | HIV status disclosure | | Unadjusted analysis | +Adjusted analysis | HIV status disclosure | | PSM analysis |
| Outcome | Level | No | Yes | OR (95% CI) | aOR (95% CI) | No | Yes | OR (95% CI) |
|---|---|---|---|---|---|---|---|---|
| Adherence to clinic visits in the past 6 months | No | 511 (76.7) | 164 (56.0) | 1 | 1 | 171 (68.7) | 145 (58.2) | 1 |
| | Yes | 155 (23.3) | 129 (44.0) | 2.59 *** (1.94, 3.48) | 2.04 ** (1.43, 2.91) | 78 (31.3) | 104 (41.8) | 1.63** (1.13, 2.36) |
| Patient representation in the past 6 months | No | 132 (19.8) | 81 (27.6) | 1 | 1 | 47 (18.9) | 71 (28.5) | 1 |
| | Yes | 534 (80.2) | 212 (72.4) | 0.65 *** (0.47, 0.89) | 0.55 *** (0.39, 0.79) | 202 (81.1) | 178 (71.5) | 0.49** (0.32, 0.76) |

Adjusted analysis included all matching covariates; 2) Statistical significance codes:

*** p<0.001,

** p< 0.01,

* p<0.05.

+: Adjusted for age, sex, marital status, level of education, source of income, distance from home to a health facility, alcohol consumption, smoking status, duration on ART, pre-ART counseling, pre-tuberculosis preventive therapy counseling, and tuberculosis preventive therapy-related side effects.

of 2.2 was needed to achieve a statistically significant lower bound of 0.04, a point at which hidden bias is evident or where propensity-score matched analysis fails to remove confounding. Since a large increase in gamma value is needed to achieve statistical significance, this indicates that our results are robust to unobserved confounders and the matching approach.

## Discussion

We evaluated the effect of disclosure of HIV status on adherence to HIV clinic visits and patient representation. We found that disclosure of HIV status is associated with a higher likelihood of adherence to clinic visits and a lower likelihood of patient representation. Adherence to clinic visits is an important component of HIV clinical care because it enables HIV care providers to follow-up on the patient's clinical progress, presents an opportunity for the patient to receive valuable information about their health from the attending healthcare provider, and is an indirect measure of the patient's attitude towards health care. Adherence to clinic visits is important because previous studies have shown that it is associated with good drug adherence and clinical progress [33]. In other studies, missed appointments are associated with detectable viral load and poor immune recovery [34, 35]. Adherence to clinic visits is a proxy measure for adherence to ART [36, 37] because missed visits translate to a lack of medicines. In a recent Ghanaian study, Lokpo et al. (2020) observed statistically significant differences in viral suppression and adherence to clinic visits. The authors found that all the participants with detectable viral load had not adhered to clinic visits while those with undetectable viral load had adhered to clinic visits in the past 12 months [36]. These findings underscore the importance of adherence to clinic visits and low patient representation in achieving better outcomes of ART. Disclosure of HIV status might have mediated the association between HIV status disclosure and better outcomes of ART, namely undetectable viral load, immune recovery, and improvements in clinical status.

Although patient representation is one of the forms of clinic attendance, it has been associated with poor adherence to ART [38], and among hypertensive patients, the practice is neither associated with adherence to medications nor effective blood pressure control [39].

The lower patient representation among patients that have disclosed HIV status is a positive outcome of the disclosure. Individual patient attendance rather than patient representation is advantageous as it enables the uptake of ongoing psychosocial support, access to clinical

reviews, laboratory and clinical monitoring for response to ART, and assessment of adherence to ART among others. Our findings of improved adherence to clinic visits and reduced patient representation could be explained by several factors. Disclosure of HIV status is encouraged among people living with HIV and they are supported to do so through ongoing psychosocial support. Therefore, it is likely that participants with disclosed HIV status have better social support, more control over their health, and are well prepared to face the challenges associated with disclosure of HIV status at both household and community levels.

Although our data show that disclosure of HIV status reduces patient representation, it should be noted that disclosure of HIV status is not mandatory in Uganda. Second, there is a possibility that some patients might have disclosed their HIV status to the representatives but we do not have sufficient data to support this claim. This should be a subject for further research.

## Methodological implications

We found similar results for the effect of disclosure of HIV status on adherence to clinic visits and patient representation for the two analytical approaches, although the measures of effect vary in magnitude. The measure of the effect for disclosure of HIV status on adherence to clinic visits in the unadjusted and adjusted analyses was attenuated in the PSM analysis. On the other hand, the effect of disclosure of HIV status was relatively lower in the unadjusted and adjusted analysis relative to the PSM analysis. These differences are likely attributable to confounding [40].

The results of the PSM analysis provide a more accurate measure of treatment effect compared to the unadjusted and adjusted logistic regression analyses.

## Study strengths and limitations

Our study has some important strengths. We used propensity-score matching and this approach enabled the estimation of unbiased effects of disclosure of HIV status since selection bias and confounding was removed. Our results are less likely to be biased by unmeasured confounders or the matching approach since sensitivity analysis showed robustness. Although our sample size reduced after the matching, the remaining sample size was relatively large to generate reliable conclusions since it met the minimum sample size for propensity-score matched analysis of 10(p+1), where p is the number of matching variables [41, 42]. However, there are limitations. We relied on secondary data which is prone to inaccuracies and missing entries. Our analysis did not include data on other confounders such as scheduling of clinic visits, participant's functional status, and comorbidities during the review period among others. We did not examine the reasons for non-disclosure of HIV status as the data analyzed was secondary. Despite the limitations, to the best of our knowledge, this is the first study in Uganda to evaluate the effect of disclosure of HIV status on adherence to clinic visits and patient representation. Our findings strengthen the practice to encourage disclosure of HIV status in HIV programming.

## Conclusions and recommendations

Our data show that disclosure of HIV status improves adherence to clinic visits and reduces patient representation among people living with HIV in eastern Uganda. We recommend that people living with HIV should be supported through ongoing counselling to disclose their HIV status to improve adherence to clinic visits and reduce representation.

## Supporting information

**S1 File. Dataset.**
(DTA)

## Acknowledgments

We thank the ART/HIV care team at Buyende Health Center for their invaluable support during the data collection. We thank the members of the Clarke International University Research Ethics Committee for their valuable comments.

## Author Contributions

**Conceptualization:** Jonathan Izudi, Paul Lwevola.

**Data curation:** Jonathan Izudi, Paul Lwevola, Damazo Kadengye, Francis Bajunirwe.

**Formal analysis:** Jonathan Izudi, Damazo Kadengye, Francis Bajunirwe.

**Investigation:** Stephen Okoboi, Paul Lwevola.

**Methodology:** Jonathan Izudi, Francis Bajunirwe.

**Project administration:** Jonathan Izudi, Paul Lwevola.

**Software:** Jonathan Izudi.

**Supervision:** Damazo Kadengye, Francis Bajunirwe.

**Validation:** Jonathan Izudi, Stephen Okoboi, Damazo Kadengye, Francis Bajunirwe.

**Visualization:** Jonathan Izudi, Stephen Okoboi, Damazo Kadengye, Francis Bajunirwe.

**Writing – original draft:** Jonathan Izudi, Stephen Okoboi, Damazo Kadengye, Francis Bajunirwe.

**Writing – review & editing:** Jonathan Izudi, Stephen Okoboi, Damazo Kadengye, Francis Bajunirwe.

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
