## [Decision Letter · Decision Letter 0]

31 Jul 2021

PONE-D-21-12455

Effect of disclosure of HIV status on patient representation and adherence to clinic visits in eastern Uganda: a propensity-score matched analysis.

PLOS ONE

Dear Dr. Izudi,

Thank you for submitting your manuscript to PLOS ONE. After careful consideration, we feel that it has merit but does not fully meet PLOS ONE’s publication criteria as it currently stands. Therefore, we invite you to submit a revised version of the manuscript that addresses the points raised during the review process.

Both reviewers consider your analysis to have been conducted appropriately and have no major concerns. However there are important clarifications in the methods needed to make the manuscript more suitable for publication, in addition to some minor concerns in the discussion. 

We look forward to receiving your revised manuscript.

Kind regards,

Deborah Donnell, Ph. D.

Academic Editor

PLOS ONE

Journal Requirements:

Reviewers' comments:

Reviewer's Responses to Questions

**Comments to the Author**

1. Is the manuscript technically sound, and do the data support the conclusions?

Reviewer #1: Yes

Reviewer #2: Yes

2. Has the statistical analysis been performed appropriately and rigorously? 

Reviewer #1: Yes

Reviewer #2: Yes

3. Have the authors made all data underlying the findings in their manuscript fully available?

Reviewer #1: Yes

Reviewer #2: Yes

4. Is the manuscript presented in an intelligible fashion and written in standard English?

Reviewer #1: Yes

Reviewer #2: Yes

5. Review Comments to the Author

Reviewer #1: The study did a good job demonstrating the relationship between HIV disclosure and adherence to clinic visits/patient representation using propensity score matching. The topic is well-covered and the result is solid. I do, however, have concerns about the method description. The participants, study design, and study procedure need more clarification.

I elaborate on my minor concerns below in detail:

Methods:

1. Please clarify the participants selection procedure. How were the 959 participants allocated/selected? What is the relationship between them and the “60,000 people” mentioned in page 5, line 102?

2. What is the point of describing “patients who are stable on ART” and “who are not” (page 5, line 109-115)? Are those inclusion/exclusion criteria of the participants? If so, please state it explicitly. If not, please provide the inclusion/exclusion criteria.

3. Please rewrite the first paragraph of “study design” (page 6, line 139-145) because the study is not an intervention study and has nothing to do with RCT. Also, instead of “intervention vs. comparison groups” (page 7, line 156) or “treated vs. untreated groups” (page 13, line 239), “disclosed vs. non-disclosed group” would be better.

4. Please clarify the measurements of key variables (i.e., disclosure, patient representation, adherence to clinic visits) in bullet points. Use one additional line(s) listing the covariates. Page 7, line 161-167 are definitions rather than measurements. Measurements should be more specific and operative. For example, is patient representation continuous (e.g., how many times did the patient send a representative in the past 6 months) or dichotomous (e.g., did the patient send a representative at least once in the past 6 months)?

5. All participants seemed to have at least 6 months history of clinic visits. Did all participants have the same treatment regimen (e.g., all of them were recommended to follow-up every month)? Add the description in “the operation of the ART clinic” section.

6. Disclosure status was recorded at week 2. Would it be updated in the follow-ups? If not, please add its potential impact on your results in the discussion section.

Results:

7. Page 14, line 260. Why were there 534 participants who did not disclose but did send representatives? Intuitively, participants who sent representatives must be already disclosed because at least they have to tell their representatives.

Discussion:

8. Page 17, line 305. Since “patient representation is one of the forms of clinic attendance,” does patient representation count in the “adherence to clinic visits” variable? Clarify the measurement of “adherence to clinic visits” in the measurements section.

9. Page 18, line 315. I highly doubt that “participants who disclosed suffer less stigma” because considerable evidence shows the opposite. Disclosure is encouraged but “reducing stigma” is not one of the reasons. You may say “disclosure is a step to social support,” which is well-established.

Reviewer #2: The authors used propensity score matching to explore the effect of disclosure of HIV status on patient representation and adherence to clinic visits in eastern Uganda. The article is well constructed, and the analysis was well performed.

• The time of the study should be reported in the abstract and the methods section.

• Quasi-experimental designs are different from observational studies and have specific characteristics. Sometimes propensity-matched designs are called quasi-randomised studies. It would be better not to use quasi-experimental for the type of this study.

• “Exposure group” should be used instead of the “intervention group”. Disclosure by a patient is not an intervention.

• What does mean treatment in the following sentences: “These covariates were selected because they are known to be associated with the study outcomes and treatment assignment thus preserving the assumption of conditional independence or strong ignorability of treatment assignment.”

• The details of the matching method should be reported (matching without replacement or matching with replacement, greedy or optimal matching, one-to-one pair matching or many-to-one (M: l) matching).

• What was the rationale for choose 0.0265 as the calliper?

• What was the type of estimated effect, ATT or ATE?

• Int Table1 “level of education” and “distance to a health facility” also have SMD>0.2 and should be marked with *.

6. PLOS authors have the option to publish the peer review history of their article (what does this mean?). If published, this will include your full peer review and any attached files.

Reviewer #1: **Yes: **Tianyue Mi

Reviewer #2: No

---

## [Author Response · Author response to Decision Letter 0]

10 Aug 2021

Reviewer #1:

 The study did a good job demonstrating the relationship between HIV disclosure and adherence to clinic visits/patient representation using propensity score matching. The topic is well-covered and the result is solid. I do, however, have concerns about the method description. The participants, study design, and study procedure need more clarification. I elaborate on my minor concerns below in detail:

Methods:

1. Please clarify the participant’s selection procedure. How were the 959 participants allocated/selected? What is the relationship between them and the “60,000 people” mentioned in page 5, line 102?

Response: We have clarified the selection procedure and it reads: “The study population consisted of a census of PLHIV started on ART between October 2018 and September 2019.”

Please note that the “60,000 people” mentioned in the study setting is the total population in the catchment area for the health facility which we thought it important to enable readers under the context within which the study was done.

2. What is the point of describing “patients who are stable on ART” and “who are not” (page 5, line 109-115)? Are those inclusion/exclusion criteria of the participants? If so, please state it explicitly. If not, please provide the inclusion/exclusion criteria.

Response: Thank you for this comment. We have included a new heading marked “Study population”. Under this heading, we have clearly described the study population including the inclusion and exclusion criteria. The sentence reads as follows: “The eligible participants included those aged ≥15 years and enrolled in care for ≥6 months during the review period. We excluded participants transferred to other health facilities because it was logistically infeasible to follow all of them and obtain data about their HIV disclosure status. We also excluded participants who were documented dead to prevent a biased estimate of the HIV disclosure effect. Further, we excluded participants whose disclosure of HIV status had occurred after the study outcomes as this would result into an inaccurate measure of the temporal relationship between disclosure of HIV status and the study outcomes.”

3. Please rewrite the first paragraph of “study design” (page 6, line 139-145) because the study is not an intervention study and has nothing to do with RCT. Also, instead of “intervention vs. comparison groups” (page 7, line 156) or “treated vs. untreated groups” (page 13, line 239), “disclosed vs. non-disclosed group” would be better.

Response: Thank you for this comment. We have used “disclosed HIV status” for the exposed group and “undisclosed HIV status” for the unexposed group. We have removed the use of intervention versus comparison. 

4. Please clarify the measurements of key variables (i.e., disclosure, patient representation, adherence to clinic visits) in bullet points. Use one additional line(s) listing the covariates. Page 7, line 161-167 are definitions rather than measurements. Measurements should be more specific and operative. For example, is patient representation continuous (e.g., how many times did the patient send a representative in the past 6 months) or dichotomous (e.g., did the patient send a representative at least once in the past 6 months)?

Response: Thank you for the comments. We had defined the exposure (disclosure of HIV status) in the earlier submission as follows: “Disclosure of HIV status was defined as revealing HIV positive status to a sexual partner(s), family members, or others in the social circle”. 

For the study outcomes, we have made revisions and the revised definitions reads as follows: 

• Patient representation: This was measured as the practice where the individual patient does not show up at the HIV clinic on the scheduled date but delegates someone to pick up the medications for them. Participants who did not show up at the HIV clinic on one or more occasions in the past 6 months were considered to have been represented. This outcome was measure as a binary variable (yes or no).

• Adherence to clinic visits: This was a binary variable (yes or no) measured as adherence to the visit at the HIV clinic where the individual patient attends clinic on the date that he/she was scheduled or within seven days before or after the scheduled date. All participants who did not adhere to their scheduled visit within the recommended period (±7 days) were considered non-adherent to the clinic visits.

The covariates were described as follows:

• Matching covariates: These included age in years (≤24, 25-50, and >50), sex (female or male), marital status (single, married, and separated), level of education (none, primary, secondary, tertiary, and above), availability of a source of income (no or yes), estimated distance from home to a health facility in kilometers (<5, 5-10, >10), ease of access to health facility (no or yes), current alcohol consumption (no or yes), current smoking status (no or yes), duration on ART in years (<5, 5-9, >10), receipt of pre-ART counseling (no or yes), receipt of pre-tuberculosis preventive therapy counseling (no or yes), and experience of tuberculosis preventive therapy-related side effects (no or yes).

5. All participants seemed to have at least 6 months history of clinic visits. Did all participants have the same treatment regimen (e.g., all of them were recommended to follow-up every month)? Add the description in “the operation of the ART clinic” section.

Response: Thank you for this comment. We enrolled participants who were in care for 6 months and beyond and this eligibility criteria has been described in response to comment #2. 

Regarding the regimen, the participants had different treatment regimens and we have added the following sentence in the study setting section for clarity: “Although the study participants have different ART regimens, the clinic uses an appointment system that enables multi-month dispensing of drugs, usually is 1-2 months of refill.”

6. Disclosure status was recorded at week 2. Would it be updated in the follow-ups? If not, please add its potential impact on your results in the discussion section.

Response: Under the "measurements section", we had earlier described the updating of disclosure status and the sentence reads: "The exposure in this study was the disclosure of HIV status measured as a dichotomous variable at the second visit (week 2) after initiation of ART and updated as treatment progresses."

Results:

7. Page 14, line 260. Why were there 534 participants who did not disclose but did send representatives? Intuitively, participants who sent representatives must be already disclosed because at least they have to tell their representatives.

Response: We have highlighted this issue in the discussing patient representation. The new sentence reads: 

“Although our data show that disclosure of HIV status reduces patient representation, it should be noted that disclosure of HIV status is not mandatory in Uganda. Second, there is a possibility that some patients might have disclosed their HIV status to the representatives but we do not have sufficient data to support this claim. This should be a subject for further research.”

Discussion:

8. Page 17, line 305. Since “patient representation is one of the forms of clinic attendance,” does patient representation count in the “adherence to clinic visits” variable? Clarify the measurement of “adherence to clinic visits” in the measurements section.

Response: Thank you. We have clarified the measurement of “adherence to clinic visit”. Please refer to the response to comment #4.

9. Page 18, line 315. I highly doubt that “participants who disclosed suffer less stigma” because considerable evidence shows the opposite. Disclosure is encouraged but “reducing stigma” is not one of the reasons. You may say “disclosure is a step to social support,” which is well-established.

Response: We have revised the sentence and it now reads: “Therefore, it is likely that participants with disclosed HIV status have better social support, more control over their health, and are well prepared to face the challenges associated with disclosure of HIV status at both household and community levels.”

Reviewer #2: 

1. The authors used propensity score matching to explore the effect of disclosure of HIV status on patient representation and adherence to clinic visits in eastern Uganda. The article is well constructed, and the analysis was well performed.

Response: Thank you for this comment.

2. The time of the study should be reported in the abstract and the methods section.

Response: We have stated the time frame for the data collection as follows:

• In the abstract section, we stated “….we performed a propensity-score-matched analysis on observational data collected between October 2018 and September 2019….” 

• In the methods section, we stated “The study population consisted of a census of PLHIV started on ART between October 2018 and September 2019.”

3. Quasi-experimental designs are different from observational studies and have specific characteristics. Sometimes propensity-matched designs are called quasi-randomised studies. It would be better not to use quasi-experimental for the type of this study.

Response: We have changed the study design from “quasi-experimental” to “quasi-randomized”.

4. “Exposure group” should be used instead of the “intervention group”. Disclosure by a patient is not an intervention.

Response: Thank you for this comment. This issue was equally raised by reviewer #1 in comment #3. We have used “disclosed HIV status” and “undisclosed HIV status” to imply “exposed” versus “unexposed” groups, respectively.

5. What does mean treatment in the following sentences: “These covariates were selected because they are known to be associated with the study outcomes and treatment assignment thus preserving the assumption of conditional independence or strong ignorability of treatment assignment.”

Response: The purpose of propensity-score matching is to reduce confounding and to do so, the assumption for the absence of confounding or unfoundedness has to be met. 

To ensure clarity, we have improved the sentence and it reads: “These covariates were selected because they are known to be associated with the study outcomes and exposure thus preserving the assumption of unconfoundedness of the association between the exposure and the outcome(s). 

6. The details of the matching method should be reported (matching without replacement or matching with replacement, greedy or optimal matching, one-to-one pair matching or many-to-one (M: l) matching).

Response: we have briefly explained the details of the matching used. Below is our description: 

“We employed greedy matching approaches namely nearest neighbor matching with and without caliper adjustment. Caliper is the distance within which the matches were considered. In nearest neighbor matching without caliper adjustment, one participant in the HIV disclosed group was selected at random and matched to one participant in the undisclosed HIV status with the closest propensity score. In nearest neighbor matching with caliper adjustment, the matching was performed within a caliper of 20% of the standard deviation of the propensity score to prevent bias from distant matches. The matching was performed without replacement. We also employed optimal matching namely optimal pair matching and optimal full matching. In optimal full matching, participants with disclosed HIV status were matched to those with undisclosed HIV status in the ratio of 1: many or many: 1. In addition, we performed exact matching where the participants were matched on the same value of propensity score.”

7. What was the rationale for choose 0.0265 as the calliper?

Response: We have explained the meaning of caliper under the matching approaches in response to comment #6. We have improved the sentence for clarity to read as: “The caliper used was 0.0265, which was 20% of the standard deviation of the propensity score.”

8. What was the type of estimated effect, ATT or ATE?

Response: The type of estimated effect was the average treatment effect on the treated or ATT. We have added the following sentence in the analysis section to ensure clarity: “The odds ratio indicates the measure of average treatment effect on the treated (ATT), a measure of the effect of HIV status disclosure for those with disclosed HIV status.” 

9. Int Table1 “level of education” and “distance to a health facility” also have SMD>0.2 and should be marked with *.

Response: This is true. We have added the asterisk (*).

---

## [Editor Report · Decision Letter 1]

5 Oct 2021

Effect of disclosure of HIV status on patient representation and adherence to clinic visits in eastern Uganda: a propensity-score matched analysis.

PONE-D-21-12455R1

Dear Dr. Izudi,

We’re pleased to inform you that your manuscript has been judged scientifically suitable for publication and will be formally accepted for publication once it meets all outstanding technical requirements.

Kind regards,

Deborah Donnell, Ph. D.

Academic Editor

PLOS ONE

Additional Editor Comments (optional):

Generally, the new submission is responsive to reviewers and is much improved as a result.

Teh following minor corrections are requested prior to publicaiont

1) In the abstract the outcome "patient representation" is not a generally a understood phrase. Please more explicitly state you meaning patients having other people pick up their prescriptions.

2) correct the last sentence of the intorduction to avoid "likelihood to use of condoms"

3) Ib Dta abstraction: "eligible patients" needs to be define.
---

## [Editor Report · Acceptance letter]

8 Oct 2021

PONE-D-21-12455R1 

Effect of disclosure of HIV status on patient representation and adherence to clinic visits in eastern Uganda: a propensity-score matched analysis 

Dear Dr. Izudi:

I'm pleased to inform you that your manuscript has been deemed suitable for publication in PLOS ONE. Congratulations! Your manuscript is now with our production department. 

Kind regards, 

on behalf of

Dr. Deborah Donnell 

Academic Editor

PLOS ONE